# Fitting of the In Vitro Gas Production Technique to the Study of High Concentrate Diets

**DOI:** 10.3390/ani10101935

**Published:** 2020-10-21

**Authors:** Zahia Amanzougarene, Manuel Fondevila

**Affiliations:** Departamento de Producción Animal y Ciencia de los Alimentos, Instituto Agroalimentario de Aragón (IA2), Universidad de Zaragoza-CITA, M. Servet 177, 50013 Zaragoza, Spain; zahiaagro@yahoo.fr

**Keywords:** gas production, pH, bicarbonate ion, high concentrate feeding, semicontinuous system

## Abstract

**Simple Summary:**

The in vitro gas production technique, either based on volume or pressure measurements, was initially set up for the evaluation of the rate and extent of fermentation of feeds for ruminants. Since it is carried out under pH conditions simulating a well-buffered medium (from pH 6.5 to 6.8), it has been generally focused to evaluation of forages and fibrous by-products, or by estimating fermentation of concentrate feeds (cereals, protein sources) for extrapolation of their use in mixed diets. However, it has also been used for determination of the nutritive value of feeds in all-concentrate diets, without taking into account that in such cases pH may range between 6.5 and 5.8, and often below this range, creating unfavourable conditions for bacterial fermentation. Modifying the concentration of bicarbonate ion in the incubation solution allows to adjust the incubation pH to conditions that simulate the in vitro fermentation conditions to those occurring under high-concentrate feeding. This highlights the importance of the incubation pH for the estimation of fermentation of feeds.

**Abstract:**

In vitro rumen fermentation systems are often adapted to forage feeding conditions, with pH values ranging in a range close to neutrality (between 6.5 and 7.0). Several attempts using different buffers have been made to control incubation pH in order to evaluate microbial fermentation under conditions simulating high concentrate feeding, but results have not been completely successful because of rapid exhaustion of buffering capacity. Recently, a modification of bicarbonate ion concentration in the buffer of incubation solution has been proposed, which, together with using rumen inoculum from donor ruminants given high-concentrate diets, allows for mimicking such conditions in vitro. It is important to consider that the gas volume recorded is in part directly produced from microbial fermentation of substrates, but also indirectly from the buffering capacity of the medium. Thus, the contribution of each (direct and indirect) gas source to the overall production should be estimated. Another major factor affecting fermentation is the rate of passage, but closed batch systems cannot be adapted to its consideration. Therefore, a simple semicontinuous incubation system has been developed, which studies the rate and extent of fermentation by gas production at the time it allows for controlling medium pH and rate of passage by manual replacement of incubation medium by fresh saliva without including rumen inoculum. The application of this system to studies using high concentrate feeding conditions will also be reviewed here.

## 1. The Gas Production Technique for Estimating Rumen Fermentation

The use of in vitro techniques has become a widely applied alternative to overcome the labour, cost and time expenses of in vivo trials; at the same time, they fit better into animal welfare considerations than in vivo trials [1,2]. However, these techniques must be easily reproducible and highly correlated with in vivo parameters in order to be considered [3]. Such characteristics can be reached through the maintenance of a typical rumen microbial population, the achievement of standard rates of fermentation and the capability to predict in vivo results. In the last decades, the gas production technique has been established in most laboratories as an indirect measurement of rumen microbial fermentation, applied for the nutrient evaluation of feeds and additives [4,5,6], the estimation of microbial fermentation processes [7,8] and the evaluation of the effect of antinutritional factors over microbial activity [9,10]. This method is based on the proportional relationship between microbial digestion of a given substrate and the production of volatile fatty acids (VFA) and the consequent production of gas, mostly carbon dioxide and methane, as final catabolites from fermentation or released from the bicarbonate buffer in the rumen medium [4]. With respect to other in vitro techniques, it has advantages such as the possibility to assay a large number of samples within the same incubation run, to allow for measuring successive readings in the same repetition, to avoid drawbacks of gravimetric estimation of microbial fermentation and to evaluate small amounts of sample such as feed fractions or isolated vegetal structures.

There have been several methodological proposals to measure the gas produced from fermentation, either by recording the volume of the gas actually produced [11], by recording the pressure generated by such gas produced [12] or by using automated or semi-automated systems [13,14,15,16]. The type of incubation vessels varies among these methods, being either calibrated glass syringes, serum bottles provided with a rubber septum or bottles provided with pressure valves or other automatisation systems. Number, size and total volume of vessels vary according to their methodological standards. Despite their different approaches, there is a general agreement in the proportion of substrate to be incubated (1 g/100 mL incubation solution), as it has been proven that gas production is linearly correlated with the amount of substrate [4,17]. In contrast, the volume of incubation solution varies considerably, being 100 mL/g in the pressure method [12] and ranging from 80 [18] to 150 [11] mL/g substrate in the syringes method, and from 100 to 200 mL/g among the automated methods [13,14,15,16]. The relationship between the volume of incubation solution and the total volume of the vessel is important, as it may either affect fermentation end products and rate of fermentation if it is high [19,20] or reduce accuracy of measurements if it is low [21].

## 2. Source of Inoculum and Fitting of Incubation pH

As in other in vitro systems, microbial inoculum must be diluted in order to avoid an excessive concentration of intermediary or end-products that might inhibit fermentation through a feed-back mechanism, but not so much that nutrient dilution might limit their availability for microbes. The proportion of inoculum in the incubation solution also varies widely among systems, from 0.10 in Theodorou et al. [12] and Davies et al. [15], to 0.20 in Cone et al. [14] or 0.33 in Menke et al. [11] and Cournou et al. [16]. The low range (0.10) is justified by authors to allow for a large batch of bottles in each single incubation run for applying to feed evaluation of a wide number of substrates, and it has been proven not to affect the rate or extent of gas production respect to higher inoculum proportions [13,17,22]. However, for other type of assays, such as studies of microbial metabolism, a larger proportion of inoculum (0.20 or 0.30) should be preferable.

Another factor of major concern is the source of inoculum. As has been highlighted by most authors and reviews [4,23], the composition and type of diet given to the donor animals determines its fermentative capacity and the degree it fits to the substate to be studied. Thus, the rate and extent of fermentation of a fibrous or a concentrate feed highly depends on its evaluation with inoculum from an animal fed on either a forage- or a concentrate-type diet [24]. Therefore, it is recommended to feed donor animals with a diet similar to the substrate to be incubated in vitro, or to the in vivo feeding conditions that are intended to be studied [25]. However, despite the microbial biodiversity being maintained, the effect of differences in diet composition fed to donor animals is reduced when rumen fluid is sampled immediately before feeding. The incubation pH can also be affected by the type of inoculum. It has been observed [26] that fermentation of cereal grains in a low-buffered medium (including a reduced proportion of bicarbonate ion) with inoculum from ewes given a forage (a 2:1 alfalfa hay:straw mixture, initial pH 6.73) or a 63% concentrate diet (initial pH 6.35) reached pH differences after 10 h incubation of 0.51 pH units, and rendered different volumes of gas and total volatile fatty acids (VFA) concentration between inocula, being 0.81 and a 0.56-fold higher, respectively, in the concentrate than in the forage diet.

In addition to microbial inoculum, incubation solution in most of these systems included mineral and buffer solutions. Mineral solutions supply nutrients for microbial metabolism and maintain osmotic pressure. However, considering the usual length of the incubation period, the addition of trace minerals is not necessary in those cases when rumen inoculum comes from donor animals given well balanced diets [23].

As the technique is planned to mimic the rumen environment, experimental conditions aim to fit temperature, osmotic pressure, anaerobiosis and pH to ruminal standard levels. Regarding pH, the technique is planned for evaluating feeds under conditions representing optimal fermentation conditions for fibre fermentation by rumen inoculum. Because of this, these incubation systems have been designed for maintaining the incubation pH between 6.7 and 7.0, considered as the physiological range for an active and healthy rumen. Under ruminal conditions, the contribution of bicarbonate is responsible for the maximum buffering capacity [27,28]. Because of this, the incubation solution includes different buffering solutions to maintain pH within those levels [23], which in all cases are made up mainly of bicarbonate buffer plus a minor proportion of phosphate buffer, as variations of the buffers proposed by Goering and Van Soest [29] or McDougall [30]. In the buffer used by Goering and Van Soest [29], concentration of bicarbonate ion is 110 mM and that of phosphate ion 20 mM. The buffering activity of the incubation solution is mainly established by the equilibrium between the added bicarbonate ion (HCO_3_^−^) and the CO_2_ infused to the medium for ensuring anaerobiosis, rendering H_2_CO_3_. In contrast, according to the estimations of Beuvink and Spoelstra [31], contribution by phosphate ions to total buffering capacity is much lower than bicarbonate, and diminishes with pH, from 0.18 of total buffer capacity at pH 6.9 to 0.07 at pH 6.5, and its effect is negligible at pH 6.0. Similarly, following the calculations of Kohn and Dunlap [32], the expected contribution of phosphate ions to the total buffering capacity is less than 0.06 when pH drops below 6.50.

Mould et al. [23] compared the chemical composition of the buffer solution in different systems. The concentration of bicarbonate ion is higher in those from Menke et al. [11] and Beuvink and Spoelstra [31] compared to others [12,33]. From the same initial pH of the medium, important differences in the final pH were reported after addition of up to 12 mmol propionic acid to simulate rumen fermentation, depending on the amount and composition of the buffer included [2]. Thus, the media of Menke et al. [11] and Goering and Van Soest [29] maintained pH values above 5.50, whereas those of Theodorou et al. [12] and Huntington et al. [33] recorded pH values of 5.42 and 5.29, respectively.

This bicarbonate/phosphate buffering system is suitable for the study of fermentation of fibrous feeds, maintaining incubation pH in the range between 6.5 to 7.0, considered as optimum for maximising fibrous fermentation. However, it is not well adapted to high-concentrate feeding conditions, where rumen pH is below that range and may even drop to 6.0 or 5.5, resulting from the high production of VFA and lactate by starch fermentation, as well as from the low salivation promoted by high concentrate diets [34]. Therefore, besides other sources of variation, mostly associated with the nature of the rumen inoculum [35], the estimation of the fermentation pattern of the gas production from concentrate feeds is largely biased depending on the incubation pH. Not only for concentrates, but also for some additives, medium pH determines the extent of response in microbial fermentative activity [36,37]. Thus, evaluation of concentrate feeds and microbial fermentative activity in high concentrate feeding conditions could be biased when their gas production is assessed in standard, strongly buffered conditions. Bertipaglia et al. [38] reported that the volume of gas produced from a mixed concentrate substrate after 24 h of incubation in a semicontinuous system was 0.26 lower at pH 5.8 than at pH 6.5. Similarly, a 0.50 lower gas production can be estimated from the equations proposed by Opatpatanakit et al. [39] when pH values dropped in the same pH range.

## 3. Adapting the In Vitro Model to High Concentrate Feeding Conditions

Several in vitro analytical techniques have been developed for the study of rumen fermentation, estimating the rate and extent of microbial activity from the gas produced [4,12]. Despite their simplicity, they have been applied to the evaluation of the nutritive value of feeds, with results being highly correlated with in vivo estimates [4,6]. The application conditions of such a technique are suitable to specifically mimic the fermentation pattern of high forage diets, by maintaining an incubation pH over 6.5 using a highly buffered medium. However, these conditions are not adapted to the study of fermentative processes in intensive feeding, since rumen pH drops to values below 6.0, even reaching values close to 5.5 for some hours during the day, and gas production is related with pH. Thus, it was observed that gas production in alfalfa hay as a forage was reduced in proportions of 0.06 or 0.20 when pH dropped from 6.8 to 6.5 or 6.2 [17]. A similar or even greater effect can be expected from concentrate substrates, and therefore, the estimation of their fermentation should be considerably biased. In this regard, it has been observed that gas production from a concentrate feed after 12 h at a medium pH 5.8 was 0.77 that of the same feed maintained at pH 6.5 [38]. A high correlation (*R* = 0.914, *p* < 0.001) has been reported between total gas production and medium pH from barley grain after 12 h incubation, within an incubation pH range from 6.50 to 5.50 [40].

Most studies on microbial fermentation of cereal grains and other starchy feeds to estimate their energy value through gas production have approached it under conventional incubation conditions, maintaining a pH of 6.7–6.9 [41,42,43]. The relationship (*R*^2^ = 0.81) of in vitro neutral detergent soluble fraction digestibility and in vivo ruminal starch digestibility of nine substrates observed by Tahir et al. [43] only responds to a single sample, and the coefficient was very low if this value was excluded. Opatpanatakit et al. [39] compared the gas production from several varieties of cereal grains for 7 h of incubation at a variable range of pH using MacDougall buffer, but the range of pH, although significantly related with gas production (*R*^2^ ranging from 0.93 to 0.97), was maintained over 6.5 and did not drop from those values during incubation.

Several approaches have been followed to maintain a low incubation pH, closer to values of rumen fermentation promoted by concentrates. Some continuous culture systems [44,45] may control incubation pH by the continuous infusion of bicarbonate ion in the saliva as well as of high concentrate solutions of NaOH and HCl to adjust pH to desired levels. However, the need for the continuous infusion of buffer is not compatible with the closed batch systems, and a high production of acids from fermentation may rapidly exhaust the buffering capacity of the medium. Moreover, strong acids and bases have a lower buffering capacity than the weak ones [32]. As stated in the former section, the use of phosphate buffer is not adequate to maintain a constant, low incubation pH as its capacity declines as the pH decreases from neutrality [32].

Maintaining a low incubation pH in an in vitro closed-batch system by reducing the bicarbonate concentration in the buffering solution according to the calculations by Kohn and Dunlap [32] allowed to compare the fermentation of different carbohydrate sources under conditions simulating high concentrate feeding [26,46]. In this sense, the concentration of bicarbonate ion in the incubation solution can be theoretically adjusted to approach the desired pH values (Table 1). With ground wheat as substrate, Kliem et al. [47] reported that a 0.50 reduction in the buffer resulted in a decrease of the medium pH after 24 h from 6.7 to 5.6, whereas with complete buffer concentration the fermentation pH was maintained over 6.0.

This way of adjusting incubation pH was applied to a closed batch system for a range of desired incubation pH from 6.5 to 5.5, using barley meal as substrate [41], and results are shown in Figure 1. After 12 h incubation, pH was maintained in the range of the adjusted values (±0.15 units) for media buffered to pH 6.50, 6.25 and 6.00. However, the exhaustion of the buffering capacity in the low buffered media (those formulated for adjusting to pH 5.75 and 5.50) promoted a larger (0.25 pH units) drop from 10 h incubation onwards, resulting in final pH values of 5.51 and 5.31, and challenging microbial fermentative activity. When these pH were applied, gas production from barley after 12 h decreased quadratically because the negative effect of a lower pH was not manifested from pH of 6.00 and above (Figure 2). The finding of Beuvink and Spoelstra [31] that gas release became non-linear as fermentation medium pH fell below 6.2 emphasises that buffering capacity is limited and can become exhausted if excess fermentation end-products are generated. Using the syringes method with either 30 (200 mg substrate) or 40 mL (450 g substrate) medium, Getachew et al. [3] reported that the standard buffer is exhausted when the gas volume exceeds 90 or 130 mL, respectively.

For these studies, acetic acid was used as a model of acidification as it is the most abundant VFA in rumen fermentation, despite that propionic acid is more characteristic of what may represent a high-concentrate rumen environment. Even though it has been suggested that the acidification capacity differs among VFA depending on their pKa [12], differences among acetic, propionic and butyric acids in terms of pKa are small (values of 4.76, 4.87 and 4.82, respectively), and all of them are well below the current range of rumen pH. Consequently, there are no differences in acidification capacity among the different VFA on a molar basis [31,48].

The adjustment of the buffering capacity of the medium to fit the incubation pH has also been applied to the study of the acidification potential of cereal grains and carbohydrates of different characteristics [26,49], by allowing to ferment substrates in a low buffered medium (calculated for adjusting to pH 5.50). In these studies, the incubation period was shortened to 10 h to avoid biased results in the fermentation profile because of extreme low pH (around pH 5.0 or below) after buffer exhaustion.

In addition to medium pH, the source of rumen fluid has an important role on the pattern of in vitro fermentation [26,50] in terms of adapting nutrient evaluation of concentrate substrates to low pH rumen conditions. Rumen microbiota promoted by high forage/fibre diets is not well suited for fermenting non-fibrous carbohydrates, producing a lower extent of substrate digestion than that promoted by the microbiota from concentrate diets in terms of gas production, volatile fatty acids production and substrate disappearance [24,26,50,51]. Therefore, the choosing of donor animals’ diet is important when concentrate feeds are evaluated for ruminants. Despite that the pH of an inoculum from donor animals given a concentrate diet was lower than that from animals fed on forages [24,26], the microbial population promoted by a concentrate diet was more favourable for fermentation of non-fibrous carbohydrates than that induced by a fibrous diet. This has been previously stated in classic revisions [4,23] and has been attributed to the lack of adaptation of microbiota from a forage inoculum to the fermentation of starch and sugar substrates [26,52] and its higher pH is related to the lower extent of fermentation, with the help of the inherent buffering capacity of forage legumes when these are included in diet.

## 4. Origin of Gas and Partition into Direct and Indirect Gas

Gas production from fermentation of feed nutrients other than carbohydrates is comparatively small. A negative correlation of crude protein with respect to gas production has been reported [53,54]. This is partly because the fermentable energy of protein is about 0.30 that of carbohydrates [3,55], but also by an underestimation of protein fermentation because of the ammonia concentration in the medium. In this way, Menke and Steingass [4] indicate that ammonia resulting from amino acid digestion partly reacts with carbon dioxide giving ammonium bicarbonate, and this sequestering of CO_2_ results in a lower amount of gas recorded. Cone and van Gelder [55] observed that gas production after 40 h incubation of casein as a source of protein diminishes in 2.5 mL for every percentual unit of casein, and a higher relationship (9.8 mL per percentual unit of protein) was also observed by González Ronquillo et al. [54], related to differences in the natural content of protein in a forage grass.

The contribution of fat to gas production is negligible: when coconut oil, palm kernel oil and/or soybean oil were incubated, only 2 to 3 mL/g OM of gas were produced, while a similar amount of casein and cellulose produced about 23 and 80 mL of gas [3,4]. Gas is derived from the glycerol released from hydrolysis of triglycerides, since degradation of long-chain fatty acids to CO_2_ and VFA was less than 0.01 when incubated with rumen microorganisms in vitro [56]. Moreover, it is assumed that fats, especially unsaturated, at or over a 0.05 proportion of in forage or mixed diets may reduce rumen microbial fermentation of dry matter disappearance and nutrients such as fibre [57,58].

As stated by Beuvink and Spoelstra [31], the potential of using the time evolution of the volume of gas produced in vitro as an index of microbial fermentation is based on the proportional relationship between the microbial digestion of substrate, the production of volatile fatty acids and the volume of gas produced from the end products of microbial fermentation (mostly CO_2_ and CH_4_, direct gas), plus the release of CO_2_ by the reaction between the acids produced in fermentation and the bicarbonate buffer of the medium (indirect gas). Gas production is basically the result of fermentation of carbohydrates to acetate, propionate and butyrate, and a highly significant correlation has been observed between short chain fatty acids and gas production [59,60]. The amount of gas released from carbohydrates depends on their chemical nature, which determines the metabolic pathway, and the relationship with other feed components such as lignin, protein and tannins, that might modulate the rate and extent of fermentation by partially masking the carbohydrate structures. The molar proportions of the major VFAs produced depends on the type of substrate [31,59], rapidly fermentable carbohydrates yielding relatively higher propionate as compared to acetate, and the contrary takes place when slowly fermentable carbohydrates are incubated [3]. The following reactions show the stoichiometry of carbohydrates (hexose) fermentation and volatile fatty acid production in the rumen [61,62]:

Hexose + 2NAD + 2ADP → 2pyruvate + 2NADH + 2ATP

2pyruvate + 2 ADP + 2H_2_O → 2acetate + 2 CO_2_ + 2H_2_ + 2ATP

2pyruvate + 4NADH + 2ADP → 2propionate + 2H_2_O + 4NAD + 2ATP

2pyruvate + 2NADH + 1ADP → 1butyrate + 2H_2_ + 2CO_2_ + 2NAD + 1ATP

CO_2_ + 4H_2_ → CH_4_ + 2H_2_O + 1ATP

Methane production is necessary for preventing hydrogen accumulation in the rumen that should slow down the rate of fermentation. In any case, the partition between CO_2_ and CH_4_ does not affect the total gas volume produced, as they substitute mole by mole. Production of CO_2_, either as direct or indirect gas, from each VFA is summarised in Table 2. The gas is produced mainly when the substrate is fermented to acetate and butyrate, either from a direct or indirect origin, resulting in four and three moles of CO_2_ per mole of hexose, respectively. Substrate fermentation to propionate yields gas only from buffering of the acid, and therefore, lower gas production is associated with propionate production (two moles of CO_2_). Hence, fermentation of feeds that produce higher proportions of propionate (such as sugars, starches and other rapidly fermentable carbohydrates) is not considered at the same level than that rendering acetate or butyrate, and thus fermentation of sugars and starches, can be considered as underestimated by gas production [63]. However, according to stoichiometric calculations, differences in gas production from a concentrate feed with rumen inocula from either a high concentrate or a forage feed should render similar gas volumes when referring to per unit of VFA. Thus, observed acetate:propionate:butyrate ratios of after 10 h incubation of barley meal with inocula from either a high concentrate or a forage diet were 61:25:14 and 68:23:09, respectively [24]; thus, estimated gas volumes should be 48.4 vs. 47.6 mL/mmol VFA, respectively, of which 22.8 and 22.0 mL are direct gas.

The in vitro gas production is highly correlated with the medium pH, either in terms of direct gas, as it comes from the microbial fermentation of substrates and thus depends on the medium conditions for microbial activity, or indirect gas, resulting from the buffering capacity of the medium. The amount of indirect gas produced per unit of acid produced should depend on the initial pH, which in turn depends on the concentration of bicarbonate ion. It is important to note that, when the buffer concentration is limited to fit pH below 6.0, the buffer effect in the incubation solution can be exhausted after about 10 h of incubation [40]. A volume of indirect gas production per unit of acid produced in a water medium without inoculum (that is, without microbial activity) has been estimated, resulting in 10.86 mL/mmol VFA [40]. This is also related to the concentration of bicarbonate ions added in the incubation medium when it is reduced to fit a pH below 6.0, as previously stated. Under these conditions, the proportion of indirect gas could be estimated at each adjusted level of pH. For example, if incubation pH is adjusted to a pH of 6.0 (that is, adding 3.9 mmol bicarbonate ion/L, see Table 1), expected indirect gas production should be added in 3.55 mL gas, whereas if medium pH is 6.50 (by adding 12.4 mol bicarbonate ion/L), the increase should be 4.73 mL. Rymer et al. [48] estimated a similar indirect gas production per unit of acid (12 to 15 mL/mmol acid) with 0.099 to 0.117 M bicarbonate ion medium. As the indirect gas production tends to diminish at a low medium pH [40] because of the reducing proportion of buffer included to get a lower pH, the comparison of substrate fermentation at low pH (6.0 or lower) could be established in terms of direct gas. However, in such a case, the production of propionic acid would not be detected and considered, since it does not render direct gas [31]. Results by Bertipaglia et al. [38] showed that acetate:propionate:butyrate proportions resulting from in vitro fermentation of concentrates do not greatly change with the incubation pH, resulting in a ratio of 57:36:8 at pH 6.5 and 57:34:9 at pH 5.8. Therefore, the stoichiometric calculation of the gas (either direct or indirect) produced under similar conditions should not be greatly affected when incubation pH is reduced. In any case, complementing obtained gas production results with total concentration and molar proportion of VFA would help to clarify the magnitude of the underestimation caused by the differences in propionate proportion.

## 5. Adaptation of Gas Production to a Semicontinuous System for Estimating Fermentation

As stated above, the estimation of the gas production pattern in a closed batch system is a simplistic approach that allows for an easy estimation of the rumen microbial fermentation at a level of accuracy that makes an acceptable approach to the rate and extent of fermentation of feeds. However, other functioning factors such as the flux of input/output of solid and liquid fractions, and thus the rate of passage, also affect overall degradation, especially for the estimation of the nutritive value of concentrates or the effect of high concentrate feeding conditions. It is not the focus of this paper to describe functioning characteristics, advantages and disadvantages of widely-used rumen continuous or semicontinuous simulation procedures [44,45,64,65] and how they fit to high concentrate feeding conditions. Instead, we would like to briefly describe the adaptation of the gas production measurement as an index of microbial fermentation to a semicontinuous incubation system, in contrast to the former methods that estimate microbial activity through substrate disappearance, and how incubation pH can be adapted for simulating high concentrate feeding conditions.

Briefly, the system proposed by Fondevila and Pérez-Espés [17], and later adapted by Prates et al. [66], is a simple system composed by several (12 to 16) Erlenmeyer flasks of 120 mL total volume as incubation vessels, which are provided with an airtight two-way plug. One of the mouths has a 150 µm filter fitted at the extreme and is used for liquid input/output, simulating the average particle size for leaving the reticulorumen. The other mouth is used for fitting the manometer probe for gas measurements, allowing for an estimation of the microbial fermentation pattern. The liquid medium is extracted manually at certain time intervals, following a discrete protocol (every two hours during the first 12 h incubation, and every four hours from 12 to 24 h) and extracted samples can be processed or used for analysis of fermentation parameters. Liquid medium samples are immediately substituted by the same volume of fresh incubation solution (without rumen inoculum), stored anaerobically and at 39 °C, to maintain the same incubation volume, adjusting the volume to the desired outflow rate.

In this semicontinuous system, Bertipaglia et al. [38] adapted the incubation pH to maintain a fixed value of 5.8 for the study of high concentrate diets. The maintenance of a low pH was successfully reached by the periodic infusion of replacing incubation solution in which the concentration of bicarbonate ion was reduced. However, mimicking the daily rumen pH pattern in high concentrate feeding conditions requires an adaptation to a circadian rhythm, which consisted of a previous drop to pH 6.0-5.6 in the first 6-8 h after substrate availability, followed by a progressive recover to initial levels (pH 6.4-6.6) after 24 h [34]. Amanzougarene et al. [24] also succeed in mimicking such pH pattern by inoculating incubation solution with a minimal bicarbonate ion concentration (0.006 M, adjusting for a medium pH of 5.5) for the first 6 h, allowing for a free drop of pH, and thereafter, an incubation solution with 0.058 M bicarbonate ion to allow pH to rise to 6.5 from 8 to 24 h (Figure 3). Thus, with this simple semicontinuous approach, incubation conditions can be more closely adapted to high concentration feeding conditions, and the fermentation pattern can be monitored through the gas production technique.

## 6. Conclusions

The gas production methodology can be applied to the study of concentrates, either for feed evaluation or to compare substrate fermentation under environmental conditions simulating high concentrate feeding diets for ruminants. However, for a better accuracy, some basic characteristics of the system, such as the origin of the inoculum and the medium pH must be previously adapted. The volume of gas produced under low medium pH (below 6.2) conditions differ from that of optimum incubation pH (6.7–6.9) because of both a reduced microbial activity because of a less favourable environment and a lower contribution of the indirect gas to total production promoted by the lower concentration of bicarbonate buffer in the incubation medium.

In order to overcome the methodological limitations promoted by the simplicity of the closed batch cultures, the estimation of microbial fermentation from the volume of gas produced can also be adapted to a semicontinuous system, which allows for a better fit of incubation conditions in terms of control of liquid turnover and establishment of a desired pH pattern.

## Figures and Tables

**Figure 1 animals-10-01935-f001:**
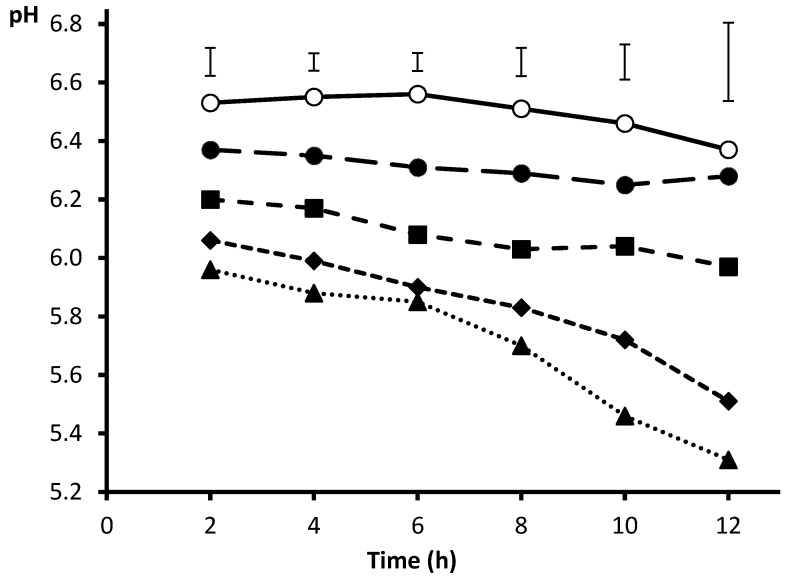
Pattern of incubation pH according to the concentration of bicarbonate ion in the medium for buffering to 6.5 (○), 6.25 (●), 6.00 (■), 5.75 (◆) and 5.50 (▲). From Amanzougarene and Fondevila [40].

**Figure 2 animals-10-01935-f002:**
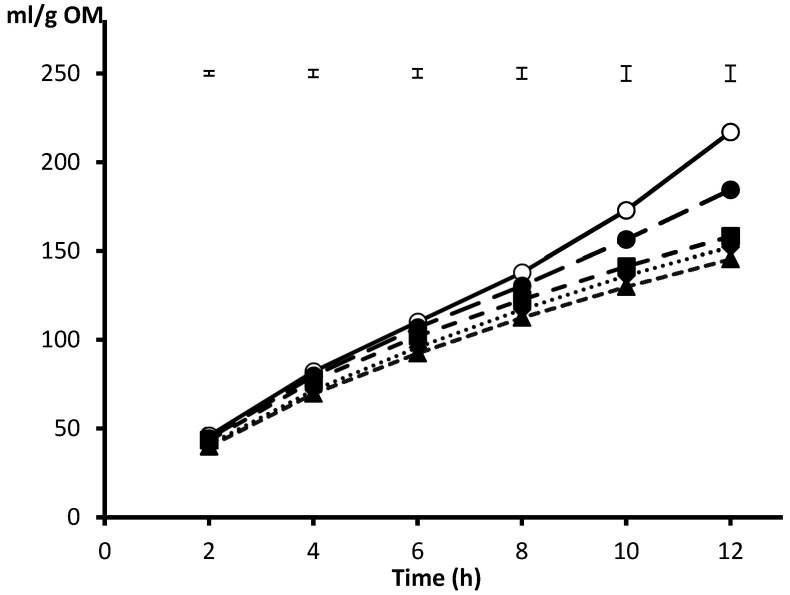
Gas production pattern from barley incubated in media buffered to 6.5 (○), 6.25 (●), 6.00 (■), 5.75 (◆) and 5.50 (▲). From Amanzougarene and Fondevila [40].

**Figure 3 animals-10-01935-f003:**
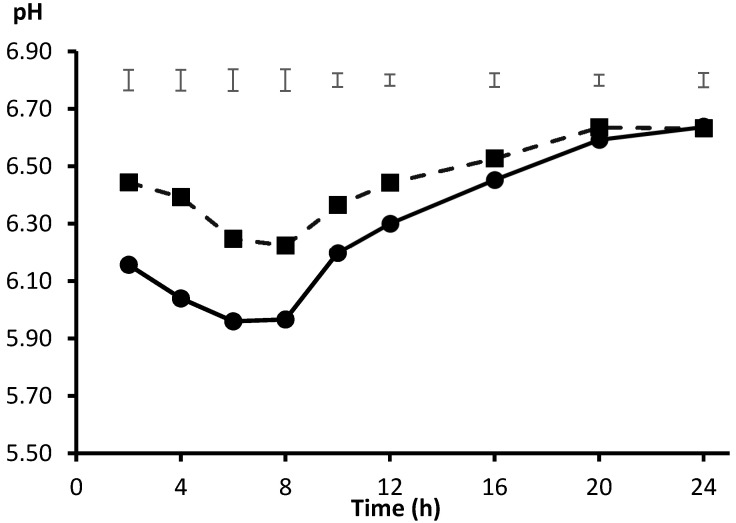
Average pattern of incubation pH from substrates incubated with inocula from a concentrate diet (CI, ●; initial pH 6.45 ± 0.15) or a forage diet (FI, ■; initial pH 6.87 ± 0.02). Upper bars show standard error of means (*n* = 3). Adapted from Amanzougarene et al. [24].

**Table 1 animals-10-01935-t001:** Estimated concentration of bicarbonate ion in the incubation solution to fit medium pH to a fixed value and final concentration (mol/L) in the medium for in vitro rumen studies (from calculations by Kohn and Dunlap [32]).

Adjusted pH Value	HCO_3_^−^ in Buffer (mol)	Buffer Composition (NaHCO_3_ + NH_4_HCO_3_, g/L)	HCO_3_^−^ in Total Medium (mol/L)
6.80	0.111	35.00 + 4.00	0.0238
6.50	0.058	18.30 + 1.90	0.0124
6.25	0.032	10.30 + 1.07	0.0068
6.00	0.018	5.70 + 0.60	0.0039
5.75	0.010	3.17 + 0.25	0.0021
5.50	0.006	1.91 + 0.12	0.0013

Medium prepared with 214 mL of buffer solution (35 g/L sodium bicarbonate plus 4 g/L ammonium bicarbonate) per litre of incubation medium with 0.10 rumen inoculum [12].

**Table 2 animals-10-01935-t002:** Molar proportions (per mmol of hexose) of direct (from microbial fermentation) and indirect (from buffering of acids) gas produced in the rumen fermentation in vitro (according to Beuvink and Spoelstra [31]).

VFA, mmol	Direct Gas, mmol	Indirect Gas, mmol	Total Gas, mmol
2 acetate	2 CO_2_	2 CO_2_	4 CO_2_
2 propionate	−	2 CO_2_	2 CO_2_
1 butyrate	2 CO_2_	1 CO_2_	3 CO_2_
2 lactate	−	2 CO_2_	2 CO_2_

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
