# Peer review of "Fitting of the In Vitro Gas Production Technique to the Study of High Concentrate Diets"

_animals, 2020, doi:10.3390/ani10101935_

Round 1

Reviewer 1 Report

Lines 43-44 - You indicate that these techniques need to be reproducible and highly correlated with in vivo data which is correct.

However, you don't really provide data or references which provide this type of information relative to using this in high concentrate diets. I feel that this type of information needs to be added to help readers determined if this is a valid approach.

 If a reader wanted to utilize this approach, can you add more details on the method or a reference  where this information can be found.

 Your 2 papers that you reference can help with the above  comments. Add some of this information to the review paper.

Author Response

Thanks for your comments.

The sentence in L43-44 refers to the general requirements in order to consider gas production or any other in vitro techniques as an index of in vivo microbial fermentation. The potential biases of using it for high concentrate-based diets are introduced later, in section 2 (L100 and ss.), following the reasoning trend of text.

Apart from work that made general evaluations of the gas production technique using a wide array of substrates (forages, concentrates, byproducts and mixed feeds), there are no much direct comparisons of in vitro and in vivo results using only concentrate feeds in literature. A mention to the work by Tahir et al (2013) has been include in new version (L167-170).

Those experimental papers from our group mentioned by Reviewer 1 described the changes in methodology that allow for apply the proposal approach. In any case, they are cited to highlight differences between incubation pH or origin of inocula, but their in vitro results were not contrasted in vivo.

Reviewer 2 Report

The review animals-959149 entitled "Fitting of the in vitro gas production technique to the study of high concentrate diets" addressed an interesting topic in vitro gas production techniques and their adaptation for high concentrate diets simulation. The review has been well conceived, however, in order to enhance the readability of the manuscript some suggestions have been done, as follow: 

L40: I strongly suggest to add a brief introduction before the paragraph on the gas production technique, for introducing the reader to the topic of the manuscript. 

L90: Please delete "In our lab" It is recommended to write in the third person. 

L109, L113: Please add before the number of references the surname's author as in L118 (Mould et al.,). 

L143: Please delete "As it has been cited above". 

L174 and 184: Please see the comment in line 109. 

Author Response

Thanks for your comments.

L40: The first paragraph of this paper refers to the general use of in vitro techniques and their advantages or limitations, so introducing a previous comment to explain gas production before might not be adequate for the order of reasoning of text. Instead, a brief reference of the concept of gas production technique has been included in L50 of the actual version.

L90: Done

L109, 113, 174 and 184: Done

L143: The comment has been removed

Reviewer 3 Report

in vitro gas production technique is an interesting way to study the fermentation pattern of the diet. However, it has several gaps, that are described in the present paper. The manuscript is well written and I have just several comments:

L89:  Define Low-buffered medium, 

L151-152: rewrite. The sentence is not clear

Thus, a 0.06 or 0.20 reduction in gas production... Are these values proportions, ml, ??

L154-157: rewrite. 

what does it mean? a high correlation (R = 0.914, P < 0.001) between total gas production from barley grain after 12 h incubation and the medium pH at 6 h within a range from 6.50 to 5.50 has been reported [40].????

L158-163: what do you want to say? Explain better or delete. 

Most of the references are written before 2010, only 12 of 66 are written after 2010. it might be interesting to look for more recent references.   

Author Response

Thanks for your comments.

L89: done

L151-152: The sentence has been rewritten to indicate those values are proportions

L154-157: The sentence has been rewritten for clarification

L158-163: This sentence aims to suggest that conditions to estimate energy value of cereals in vitro could be inadequate, and thus obtained results may be biased. The sentence has been rewritten for clarification.

Regarding dates of references, as the technique was standardised in the 80´s-90´s, most key references dealing with its optimisation were published before 2010. Later publications mostly refer to the use of the technique to evaluate either the effects of secondary compounds of forages and byproducts or the use of additives in mixed diets, but without having into account the potential bias of a high incubation pH nor the evaluation of concentrate feeds, and therefore are not directly related with the topic of this review.